



# GloUTCI-M: A Global Monthly 1 km Universal Thermal Climate Index Dataset from 2000 to 2022

Zhiwei Yang[1], Jian Peng[1], Yanxu Liu[2], Song Jiang[3], Xueyan Cheng[4], Xuebang Liu[1], Jianquan Dong[5], Tiantian Hua[1], and Xiaoyu Yu[1]

[1]Laboratory for Earth Surface Processes, Ministry of Education, College of Urban and Environmental Sciences, Peking University, Beijing 100871, China
[2]State Key Laboratory of Earth Surface Processes and Resource Ecology, Faculty of Geographical Science, Beijing Normal University, Beijing 100875, China
[3]Department of Civil and Environmental Engineering, University of Illinois Urbana-Champaign, Urbana, IL 61820, USA
[4]Faculty of Environment and Natural Resources, University of Freiburg, Freiburg im Breisgau, 79106, Germany
[5]School of Landscape Architecture, Beijing Forestry University, Beijing, 100083, China

*Correspondence to*: Jian Peng (jianpeng@urban.pku.edu.cn)

**Abstract.** Climate change has precipitated recurrent extreme events and emerged as an imposing global challenge, exerting profound and far-reaching impacts on both the environment and human existence. The Universal Thermal Climate Index (UTCI), serving as an important approach to human comfort assessment, plays a pivotal role in gauging how the human adapts to meteorological conditions and copes with thermal and cold stress. However, the existing UTCI datasets still grapple with limitations in terms of data availability, hindering their effective application across diverse domains. We have produced the GloUTCI-M, a monthly UTCI dataset boasting global coverage, an extensive time series spanning from March 2000 to October 2022, and a high spatial resolution of 1km. This dataset is the product of a comprehensive approach leveraging multiple data sources and advanced machine learning models. Our findings underscore the superior predictive capabilities of CatBoost in forecasting UTCI (MAE = 0.747°C, RMSE = 0.943°C, R$^2$ = 0.994) when compared to machine learning models such as XGBoost and LightGBM. Utilizing GloUTCI-M, the geographical boundaries of cold stress and thermal stress areas on a global scale were effectively delineated. Over the span of 2001 to 2021, the mean annual global UTCI registers at 17.24°C, with a pronounced upward trend. Countries like Russia and Brazil emerge as key contributors to the mean annual global UTCI increase, while countries like China and India exert a more inhibitory influence on this trend. Furthermore, in contrast to existing UTCI datasets, GloUTCI-M excels at portraying UTCI distribution at finer spatial resolutions, augmenting data accuracy. This dataset enhances our capacity to evaluate thermal stress experienced by the human, offering substantial prospects across a wide array of applications. The GloUTCI-M is publicly available at https://doi.org/10.5281/zenodo.8310513 (Yang et al., 2023).



# 1 Introduction

Global climate change has precipitated recurrent extreme events, presenting formidable challenges to society and the environment (Tripathy et al., 2023; Pokhrel et al., 2021). These challenges encompass threats to human health, degradation of ecosystems, and heightened energy demands (Deroubaix et al., 2021; Kotcher et al., 2021; Outhwaite et al., 2022).

Temperature, being the preeminent parameter within meteorological variables, serves as an instrument for monitoring climate fluctuations and is imperative for the formulation of policies and the implementation of appropriate response measures (Yang et al., 2020; Yin et al., 2023; Peng et al., 2020b). Nevertheless, the human genuine awareness of its surroundings, denoted as human comfort, assumes greater significance in a comprehensive evaluation of the influence of environmental conditions (Gobo et al., 2022). Human-perceived cold or thermal stress is intricate, intimately linked to various meteorological variables. For instance, wind speed can either amplify or mitigate perceived body temperature, humidity can modulate the efficiency of evaporation, and solar radiation can elevate perceived temperature when exposed to sunlight (Zhang et al., 2023b; Fahad et al., 2021). Consequently, while a solitary meteorological variable, namely temperature, remains crucial, an index that amalgamates multiple meteorological variables is better poised to mirror the authentic human perception of the ambient environment.

Currently, several indices pertaining to human comfort have been widely adopted, encompassing the heat index, wet-bulb temperature, and humidity index (Vargas Zeppetello et al., 2022; Freychet et al., 2020). The Universal Thermal Climate Index (UTCI), a novel index of human comfort, excels in portraying the human responses to thermal and cold stress more accurately (Bröde et al., 2012). The UTCI hinges on the concept of equivalent temperature, defined as the temperature within a standardized reference environment, furnishes a more comprehensive and precise portrayal of the human perceptions under diverse meteorological circumstances (Bröde et al., 2012). By integrating a gamut of meteorological variables, including temperature, humidity, wind speed, and solar radiation, among others, the UTCI aptly characterizes comfort levels across varying environments (Park et al., 2014). As the advanced biometeorological index, the UTCI has objectivity in assessing the impact of the atmospheric milieu on the human organism (Zare et al., 2018). Currently, the UTCI is extensively employed in studies concerning short-term repercussions of atmospheric conditions on the human, urban bioclimatology, and evaluations of the urban heat island phenomenon (Hwang et al., 2022; Kyaw et al., 2023; Zhang et al., 2023c). Consequently, the UTCI, with its incorporation of multiple meteorological variables and hallmark objectivity and comprehensiveness, can better characterize the thermal and cold stresses experienced by humans and has gained widespread currency across numerous disciplines.

Several datasets encompassing human comfort indices have been produced for global or localized domains (Zhang et al., 2023a; Dong et al., 2022). However, there is a notable dearth of datasets dedicated to UTCI, which hinders in-depth research and the application of UTCI. The existing UTCI datasets predominantly exhibit low spatial resolutions, such as the ERA5-HEAT with a spatial granularity of 0.25° (encompassing the globe) and the HiTiSAE with a spatial granularity of 0.1° (encompassing East and South Asia) (Di Napoli et al., 2021; Yan et al., 2021). These prevailing UTCI datasets often



inadequate for urban and landscape scale investigations, given that these studies necessitate data of higher spatial resolution

to accurately capture intra-urban meteorological variations and human perceptions (Peng et al., 2021; Yang et al., 2021; Cao et al., 2022). Therefore, the development of a suite of globally accessible, long-time series and high spatial resolution UTCI datasets is imperative. This initiative will address the existing void in UTCI data availability and enhance the precision and practicability of UTCI for urban and landscape scale investigations.

     To facilitate the widespread future applications of UTCI data, we have produced GloUTCI-M, a monthly UTCI dataset

characterized by global coverage, a long-time series, and high spatial resolution. This work involves establishing a systematic process for generating and describing the UTCI dataset, relying on machine learning models that incorporate multiple covariates, as well as exploratory data analysis. Several key contents include: (1) Examining the connection between UTCI and various covariates, utilizing multiple machine learning models; (2) Employing the optimal machine learning model to produce a monthly high spatial resolution UTCI dataset that spans the entire globe, known as GloUTCI-M;

(3) Analysing the global spatiotemporal characteristics and pattern evolution of UTCI based on GloUTCI-M; (4) Comparing GloUTCI-M with existing UTCI datasets and evaluate the data availability of GloUTCI-M.

## 2 Data

### 2.1 Meteorological station data

     The global meteorological observations spanning from 2000 to 2022 are sourced from the Integrated Surface Database

(ISD). This database provided by the National Oceanic and Atmospheric Administration (NOAA) (https://www.ncei.noaa.gov, last access: 16 September 2023), amalgamates data from over 100 disparate raw data sources spanning the globe. These sources provide a comprehensive array of meteorological variables, including wind speed, temperature, dew point temperature, station pressures, current weather conditions, and visibility, among others. The ISD database spans the extensive timeframe of 1901 to 2023, and it serves multifarious research purposes, extending to

investigations into global climate change, climate modelling, and various domains within environmental science. We utilized the ISD_hourly subset from the ISD database. The majority of the data from ISD_hourly is available at a 3-hour interval, with a small number of meteorological stations providing data at a 1-hour interval.

     To expand the number of available meteorological samples, meteorological stations are not required to have meteorological observation data for all periods from 2000 to 2022. We enforced rigorous quality control measures for these

meteorological stations, exclusively retaining those with meteorological observation data available for a minimum of 25 days in each month. Consequently, the count of selected meteorological samples fluctuates annually and even monthly, exhibiting differences in spatial distribution (Fig. 1). In terms of spatial distribution, North America and Europe have more dense meteorological samples, while the East Asia also has a substantial cluster of meteorological samples meeting our criteria. Conversely, the southern hemisphere, notably Africa and South America, demonstrates a lower density of meteorological

stations that meet the threshold for data quality. Furthermore, there has been a consistent year-on-year increase in the number

of meteorological stations meeting our criteria in recent times. For instance, in October 2000, we had 5402 such samples, a number that surged to 8235 by October 2022. In total, our selection process yielded more than 2 million samples, ensuring a robust dataset for our analysis.

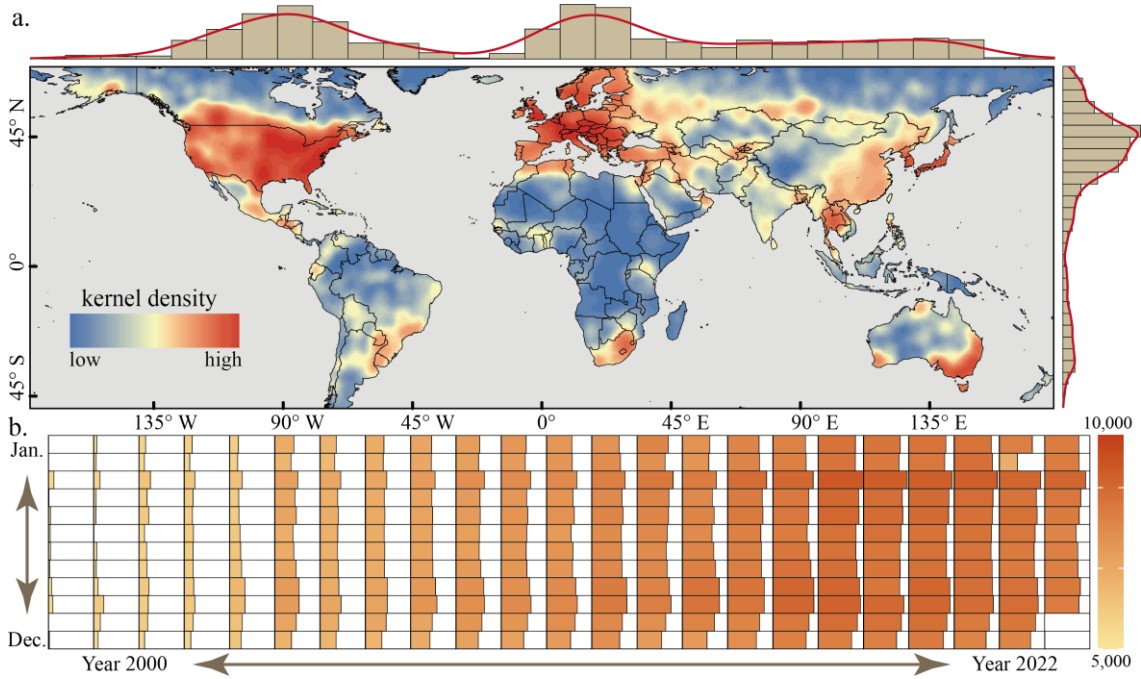

**Figure 1.** Global meteorological samples accessible for each monthly interval from 2000 to 2022: (a) kernel density distribution; (b) quantitative statistics.

## 2.2 Covariates

We opted seven key covariates in our analysis, specifically LST (Land Surface Temperature), NTL (Nighttime Lights), LULC (Land Use/Land Cover), kNDVI (kernel Normalized Difference Vegetation Index), DEM (Digital Elevation Model), Month, and LAT (latitude). Our selection of these covariates was guided by two fundamental principles. Firstly, we drew from existing research to ascertain that the chosen covariates possess a direct bearing on UTCI or wield significant influence over the meteorological variables employed in UTCI calculations (Fahad et al., 2021; Pappenberger et al., 2015; Wang et al., 2020; Peng et al., 2020a). This ensures the relevance of these covariates to our analysis. Secondly, we took care to select covariates that are openly accessible and obtainable without cost, and that originate from data sources with extensive global coverage. This step was essential to guarantee the broad applicability of our analysis, encompassing diverse regions across the world. Here are the details of our data sources and the rationale behind their selection:

LST: We acquired daytime LST data from MOD11A2 (Moderate Resolution Imaging Spectroradiometer) (https://lpdaac.usgs.gov, last access: 16 September 2023). This data source is accessible at a spatial resolution of 1 km and compiles average values over an 8-day interval from corresponding MOD11A1 LST pixels.



NTL: NTL is indicative of human activities and urbanization. We utilized NPP-VIIRS-like NTL data, available at a spatial resolution of 500 m. This dataset effectively combines data from two NTL sources (DMSP-OLS and NPP-VIIRS), extending the temporal range of NTL observations (Chen et al., 2021).

    LULC: We sourced LULC data from MODIS_IGBP Land Cover Dataset, which offers a spatial resolution of 500 m. This dataset results from supervised classification using MODIS Terra and Aqua reflectance data and categorizes land cover

into 17 distinct types (Loveland et al., 2000). It provides annual data from 2001 to 2021. In instances where we required LULC data for the years 2000 and 2022, we substituted them with data from 2001 and 2021, respectively.

    kNDVI: This novel vegetation index is calculated based on NDVI (Normalized Difference Vegetation Index), offers advantages in terms of resistance to saturation, bias, and noise (Camps-Valls et al., 2021). It also demonstrates greater stability at various spatial and temporal scales. We computed kNDVI using MOD13A2 (https://lpdaac.usgs.gov, last access:

16 September 2023).

    DEM: We employed the Multi-Error-Removed Improved-Terrain DEM (MERIT DEM) as DEM data source. This global DEM boasts high accuracy and possesses a resolution of 3 arc-seconds (approximately 90 meters at the equator) (Hirt, 2018). The MERIT DEM is generated by mitigating major error components in existing DEMs, including the NASA SRTM3 DEM and JAXA AW3D DEM.

Our pre-processing of the aforementioned covariate data involved several steps, including splicing, cropping, resampling, and monthly data synthesis. These steps were undertaken to achieve uniformity in terms of spatial extent, projection, and spatial resolution across all covariates. The covariate data were obtained via the Google Earth Engine (GEE), and resampling techniques were employed to harmonize the spatial resolution of all covariates. Specifically, we adopted the mean resampling method for covariates with continuous values and the nearest-neighbour assignment method for categorical

covariates. This ensured that all covariates have consistent spatial resolution. Furthermore, we categorized the seven covariates into two groups based on their characteristics and data availability. Dynamically evolving covariates, such as LST and kNDVI, exert a significant influence on the monthly mean value of UTCI. Therefore, we calculated their monthly mean values. Conversely, statically evolving covariates like latitude, month, and DEM impose inherent constraints on the monthly mean of UTCI. Considering temporal and spatial constraints inherent to raw covariate data availability, and to guarantee

temporal and spatial consistency across all covariate data, we ultimately derived monthly covariate data spanning from March 2000 to October 2022 within the global latitude range of 50° S to 70° N.

## 2.3 Existing UTCI datasets

    We compared the GloUTCI-M with two pre-existing UTCI datasets, namely ERA5-HEAT and HiTiSEA, to assess the accuracy of all three datasets. ERA5-HEAT stands as a worldwide historical dataset encompassing bioclimatic variables,

inclusive of UTCI (Di Napoli et al., 2021). The computation of ERA5-HEAT relies upon ERA5 reanalysis datasets, incorporating meteorological variables such as air temperature, humidity, wind speed, and more. ERA5-HEAT boasts a spatial granularity of approximately 28 km (0.25°) and a temporal resolution extending up to 1 hour. It is freely accessible

via the Copernicus Climate Data Store (https:// cds.climate.copernicus.eu, last access: 16 September 2023). The HiTiSAE is a gridded product featuring a spatial resolution of 0.1° and contains daily UTCI spanning from January 3, 1981, to December 31, 2019, for the East and South Asian regions (Yan et al., 2021).

**3 Methodology**

The production process of GloUTCI-M encompasses various key steps, such as the calculation of UTCI for meteorological samples, the acquisition of covariate data, and the identification of the optimal machine learning model (Fig. 2). We initiate by utilizing observational data to compute daily UTCI for the meteorological samples. Subsequently, we synthesize monthly UTCI based on the availability of daily UTCI data. Following this, we employ three distinct machine learning models to establish the relationship between the monthly UTCI and the covariates. Finally, we validate the accuracy of the training results generated by the machine learning models. Upon completion of this validation process, we select the optimal machine learning model to produce GloUTCI-M using the covariate raster data. Additionally, leveraging GloUTCI-M, we employ spatiotemporal analysis models, including mutation point analysis and trend analysis, to identify the spatial distribution and temporal fluctuations of global UTCI.

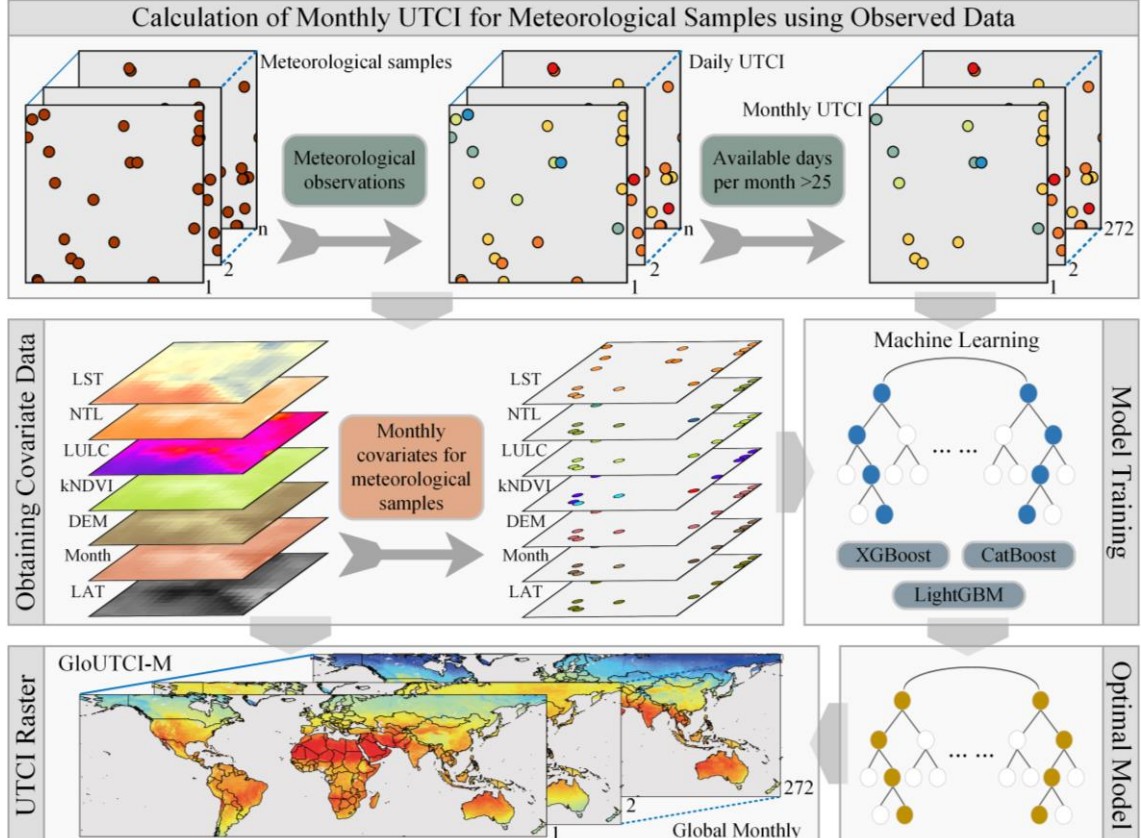

**Figure 2.** Production process of the GloUTCI-M.



### 3.1 Calculation of Universal Thermal Climate Index

The deviation between UTCI and temperature hinges on meteorological variables such as air temperature, mean radiant
temperature, wind speed, and water vapor pressure, which can be expressed by the following formula (Bröde et al., 2012).

$$UTCI = T_a + Offset(T_a, T_{mrt}, V_a, P_a) = f(T_a, T_{mrt}, V_a, P_a) \tag{1}$$

where $T_a$ represents air temperature (°C), $T_{mrt}$ denotes mean radiation temperature (°C), $V_a$ stands for wind speed (m/s),
and $P_a$ corresponds to water vapor pressure (hPa). $T_{mrt}$ and $P_a$ can be derived from data concerning humidity, solar radiation,
and solar altitude angle. Solar radiation data were sourced from ERA5-Land, solar altitude angles were calculated based on
geographical latitude and longitude coordinates. Based on available data from meteorological stations, we selected records
captured between 11:00 and 14:00 local time as the daily sample data for UTCI calculations. These daily calculations were
then employed to derive monthly UTCI for each meteorological station. The UTCI computations were executed utilizing the
BioKlima model (Bioklima ver. 2.6 Software) (https://www.igipz.pan.pl, last access: 16 September 2023).

Human comfort levels can be categorized into 10 distinct classes, predicated upon UTCI values, as follows (Bröde et al.,
2012): extreme cold stress (< -40°C); very strong cold stress (-40 ~ -27°C); strong cold stress (-27 ~ -13°C); moderate cold
stress (-13 ~ 0°C); slight cold stress (0 ~ 9°C); no thermal stress (9 ~ 26°C); moderate thermal stress (26 ~ 32°C); strong
thermal stress (32 ~ 38°C); very strong thermal stress (38 ~ 46°C); and extreme thermal stress (> 46°C). Therefore, a UTCI
value below 0°C implies the presence of cold stress, a UTCI value exceeding 26°C implies the presence of thermal stress.

### 3.2 Machine learning models

We employed a random division approach to partition the monthly UTCI and covariate data, encompassing all
meteorological samples worldwide from 2000 to 2022, into a training set (90%) and a test set for model evaluation (10%).
These subsets served as the basis for training and evaluating three prominent machine learning models: XGBoost,
LightGBM, and CatBoost.

#### 3.2.1 XGBoost

Extreme Gradient Boosting (XGBoost) is an integrated machine learning algorithm centered on decision trees and
utilizes a gradient ascent framework for classification and regression tasks (Chen and Guestrin, 2016). It excels as a tool for
massively parallel boosting trees and is characterized by its efficiency, flexibility, and portability. XGBoost employs
regularization terms in the loss function to control model complexity while approximating the loss function through a
second-order Taylor expansion, enhancing model accuracy. It employs techniques like feature subsampling, node splitting,
and handling missing values to improve model generalization. XGBoost has found wide application in machine learning
competitions and various domains, including evapotranspiration estimation, land cover classification, air quality prediction,
and aboveground biomass estimation (El Bilali et al., 2023; Katori et al., 2022). Training for this model was conducted using



the Python package *xgboost*, with hyperparameter tuning performed via grid search, encompassing all feasible combinations of hyperparameters, such as learning rate, tree depth, and the number of iterations.

### 3.2.2 LightGBM

Light Gradient Boosting Machine (LightGBM) is a boosting framework that adopts a histogram-based decision tree algorithm to enhance the computational efficiency of Gradient Boosting Decision Trees (GBDT) (Ke et al., 2017). It stands out for its faster training speed, reduced memory consumption, enhanced accuracy, and support for distributed processing. LightGBM utilizes a leaf-wise tree growth strategy to select the leaf node with the highest gain at each split, enabling faster, deeper tree growth and improving model accuracy. It employs a histogram algorithm to approximate the gradient and

Hessian matrix for enhanced accuracy. LightGBM has demonstrated its ability to expedite GBDT model training without compromising accuracy and has been applied to various prediction tasks involving spatiotemporal variables (Ahlswede et al., 2023; Aybar et al., 2022). We trained this model using the Python package *lightgbm* and conducted hyperparameter tuning through a grid search, exploring all potential hyperparameter combinations, including learning rate, tree depth, and subsampling rate.

### 3.2.3 CatBoost

Categorical Boosting (CatBoost) operates on the symmetric decision tree (oblivious trees) principle, offering few parameters, support for categorical variables, and high accuracy (Prokhorenkova et al., 2018). CatBoost introduces innovative techniques for automatically converting categorical features into numerical ones. It computes statistics on categorical features, such as category frequency, and uses hyperparameters to generate new numerical features. Moreover,

CatBoost leverages combinations of categorical features, enriching feature dimensions. The algorithm employs a sort boosting technique to tackle noisy points in the training set, mitigating Gradient Bias and addressing Prediction shift issues, which reduces overfitting and enhances model accuracy and generalization. CatBoost has seen wide application in domains like meteorology, hydrology, agriculture, and regression and prediction tasks across various fields (Tasaki et al., 2022; Cravo et al., 2022). We conducted training for this model using the Python package *catboost* and executed hyperparameter tuning

through grid search, systematically exploring all possible combinations of hyperparameters, including learning rate, number of iterations, and tree depth.

### 3.3 Model evaluation metrics

To assess the suitability of XGBoost, LightGBM, and CatBoost for predicting UTCI and determine the optimal machine learning model for producing UTCI dataset, we employed three widely recognized metrics for evaluating predictive models:

mean absolute error (MAE), root mean square error (RMSE), and the coefficient of determination ($R^2$). Both MAE and RMSE serve as metrics to gauge the overall accuracy of a model by quantifying the magnitude of the error between the predicted values and the observed values. MAE is computed as the average of the absolute differences between the predicted



and observed values, while RMSE represents the square root of the average of the squared differences between predicted and observed values. Smaller values for MAE and RMSE indicate a more precise predictive model. $R^2$ measures the extent to which the model fits the data and ranges between 0 and 1. A value closer to 1 signifies a superior model fit. The calculations for these three indicators are as follows.

$$\text{MAE} = \frac{1}{n}\sum_{i=1}^{n}|P_i - C_i| \tag{2}$$

$$\text{RMSE} = \sqrt{\frac{1}{n}\sum_{i=1}^{n}(P_i - C_i)^2} \tag{3}$$

$$R^2 = 1 - \frac{\sum_{i=1}^{n}(C_i - P_i)^2}{\sum_{i=1}^{n}(C_i - \bar{C})^2} \tag{4}$$

where $P_i$ represents the predicted UTCI, $C_i$ denotes the observed UTCI calculated based on data from the meteorological samples, $\bar{C}$ is the average value of all $C_i$, and $n$ signifies the number of samples. The test dataset was utilized to compute these evaluation metrics, thereby enabling the assessment of the machine learning model for UTCI prediction capability.

**3.4 Spatiotemporal analysis**

To comprehensively understand the spatiotemporal distribution of global UTCI, we employ two distinct analytical methods: a Bayesian model averaging time-series decomposition algorithm (BEAST) to identify the characteristics of the global UTCI from the spatial perspective, and the Theil-Sen Median analysis along with the Mann-Kendall (MK) method to understand the trend of global UTCI from a temporal perspective.

BEAST analysis is a Bayesian model averaging algorithm designed for the decomposition of numerical series data (Zhao et al., 2019). Its primary purpose is to identify key characteristics such as mutation points and nonlinear trends within the data. BEAST is advantageous because it enhances the accuracy of detecting mutation points by providing prior and posterior probability distributions for these points. It quantifies the inherent uncertainty associated with mutation detection by assigning quantitative probabilities. BEAST has been widely applied in diverse numerical series data, including financial, public health, economic, and ecological datasets (Mulverhill et al., 2023; Pitarch et al., 2021). To address the issue of potentially differing or conflicting estimation results produced by various models, BEAST employs Bayesian modelling to assess the relative importance of individual models, averages results from multiple models, and decomposes the numerical series data into three components (Zhao et al., 2019).

$$Y_\tau = T_\tau + S_\tau + \varepsilon_\tau \tag{5}$$
$$T_i = \alpha_i + \beta_i t(\tau_{i-1} < t < \tau_i, i = 1, \dots, m) \tag{6}$$

where $Y_\tau$ is the numerical series, $T_\tau$ is the trend signal, $S_\tau$ is the seasonal signal (for the time series), $\varepsilon_\tau$ is the residual signal, and $\tau$ denotes the number of mutation points. $T_i$ is the linear expression of the trend term, $i$ is the location of the mutation point, $\alpha_i$ and $\beta_i$ denote the intercept and the slope of the linear model on the two sides of the mutation point, respectively.





The trend analysis carried out by the Theil-Sen slope estimation method and the test of significance of the trend using the MK method can reflect the effective trend of each pixel in the time series. This comprehensive method has been widely used in the trend significance test of long-time series data in many fields such as ecology, meteorology and hydrology (He et al., 2022; Hu et al., 2023). The Theil-Sen slope estimation method is a robust nonparametric statistical technique used for trend calculation. It is insensitive to measurement errors and outlier data and is effective in handling missing value noise. This method computes slopes between pairs of data points in the time series and calculates the median of these slopes to determine the overall trend (Zheng et al., 2021).

$$\beta = \text{Median}\left(\frac{X_j - X_i}{j - i}\right), \forall j > i. \tag{7}$$

where $X_j$ and $X_i$ represent observed data points in the time series, and $\beta$ indicates the overall trend. If $\beta$ is greater than 0, it suggests an increasing trend, while $\beta$ less than 0 indicates a decreasing trend.

The MK method is a nonparametric test that does not require the measurements to follow a normal distribution. It is robust against missing values and outliers, and is used to test the significance of the time series trend as a supplement to the Theil-Sen slope estimation method. The test statistic is calculated based on the relationship between data values in the time series $X_i$ ($i = 1, 2, 3, ..., n$) (Peng et al., 2023).

$$Z = \begin{cases} \dfrac{S}{\sqrt{\text{Var}(S)}} & (S > 0); \\ 0 & (S = 0); \\ \dfrac{S + 1}{\sqrt{\text{Var}(S)}} & (S < 0). \end{cases} \tag{8}$$

$$\text{Var}(S) = \frac{n(n - 1)(2n + 5)}{18} \tag{9}$$

where $Z$ is the standardized test statistic; $S$ is the relationship between the size of $X_i$ and $X_j$ among all pairs of values ($X_j, X_{i,} j > i$) in the time series; $n$ is the number of data in the time series. When the absolute value of $Z$ exceeds certain critical values (1.65, 1.96, and 2.58), it indicates that the trend is statistically significant at confidence levels of 90%, 95%, and 99%, respectively. The 95% confidence level is commonly employed for this purpose.

## 4 Results

### 4.1 Evaluation of model performance

To identify the optimal machine learning model for achieving heightened accuracy in the global monthly UTCI dataset, we employed the test dataset to assess the performance of XGBoost, LightGBM, and CatBoost. Our evaluation process consisted of two key steps. Firstly, we juxtaposed the predicted UTCI generated by each model against the observed UTCI from meteorological samples, and then calculated accuracy metrics such as MAE, RMSE, and $R^2$ (Fig. 3). Secondly, we computed the mean of the absolute residuals (MAR) between the predicted UTCI and the observed UTCI at globally

available meteorological stations (Fig. 4). This allowed us to discern variations in the performance of the three models over global space.

The concordance between the predicted UTCI and the observed UTCI is notably strong for all three models, as evidenced by data points predominantly clustering around the 1:1 line. The positional deviation between the regression line
and the 1:1 line for lower UTCI values signifies varying degrees of overestimation in the UTCI predictions of the three models. Notably, LightGBM exhibits a relatively larger positional deviation between the regression line and the 1:1 line (Fig. 3 (b)), followed by XGBoost (Fig. 3 (a)), with CatBoost demonstrating the smallest deviation (Fig. 3 (c)). Regarding MAE, both XGBoost and LightGBM exhibit values close to 0.9°C, with a slightly smaller MAE for XGBoost. In stark contrast, CatBoost boasts a substantially lower MAE of 0.747°C, clearly surpassing the performance of XGBoost and LightGBM. The
RMSE among the three models exhibit significant disparities but maintain the same order as the MAEs, with CatBoost outperforming XGBoost and LightGBM. It is noteworthy that both XGBoost and LightGBM surpass an RMSE of 1.1°C, whereas CatBoost excels with an RMSE of under 1°C. Furthermore, all three models exhibit high values for the $R^2$. XGBoost and LightGBM closely approximate with an $R^2$ of 0.991, while CatBoost excels with the highest $R^2$, reaching 0.994. Consequently, the performance of CatBoost surpasses that of XGBoost and LightGBM in terms of MAE, RMSE, and
$R^2$.

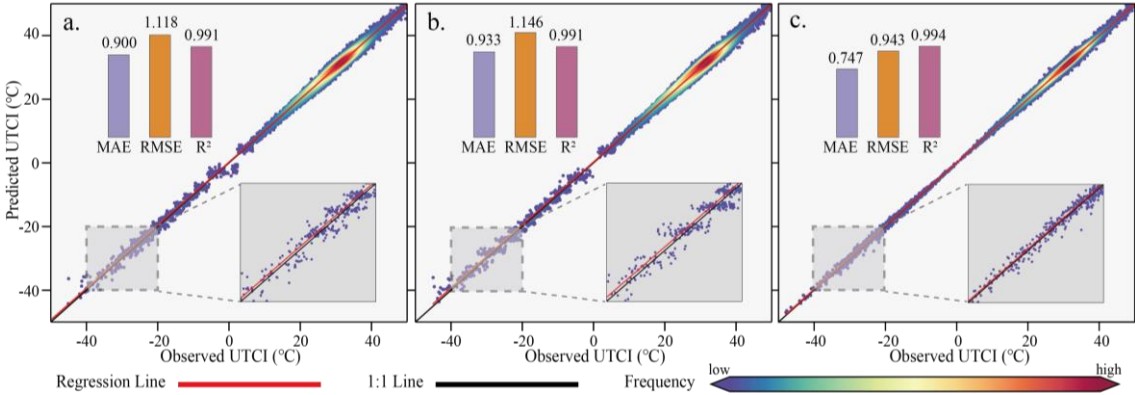

**Figure 3.** Comparison of predicted UTCI derived from machine learning models with observed UTCI obtained from meteorological samples: (a) XGBoost; (b) LightGBM; (c) CatBoost.

Utilizing the MAR between predicted UTCI and observed UTCI from all available global meteorological stations (9083
stations), we employed spatial interpolation to discern performance disparities among XGBoost (Fig. 4 (a)), LightGBM (Fig. 4 (b)), and CatBoost (Fig. 4 (c)) on a global scale. The distribution of the MAR for the three models across the global landscape exhibits a consistent pattern. Regions densely populated with meteorological stations, such as the United States and Europe, display smaller MAR, while regions with fewer meteorological stations, such as Africa and North Asia, manifest larger MAR. Notably, the MARs of XGBoost and LightGBM exhibit substantial spatial enlargement compared to
CatBoost. Specifically, MARs exceeding 1.2°C extend over significant expanses, including western Africa, the Siberian region of Russia, and northern Canada. In these regions, CatBoost's MAR remains relatively modest, with only sporadic



Data

areas where MAR exceeds 1.2°C. In other global regions, CatBoost consistently maintains a significantly lower MAR in comparison to XGBoost and LightGBM. The mean MAR for XGBoost and LightGBM is similar, standing at 0.86°C and 0.89°C, respectively, but both are notably higher than CatBoost's mean MAR of 0.49°C. Additionally, the count of stations

registering MARs exceeding 1.2°C reaches 956 for XGBoost and 1102 for LightGBM, constituting more than 10% of the meteorological stations (10.5% and 12.1%, respectively). In contrast, CatBoost displays only 96 meteorological stations with MARs exceeding 1.2°C, accounting for a mere 1.1% of the total meteorological stations. Consequently, CatBoost also demonstrates superior performance over XGBoost and LightGBM concerning the spatial and numerical distribution of MAR.

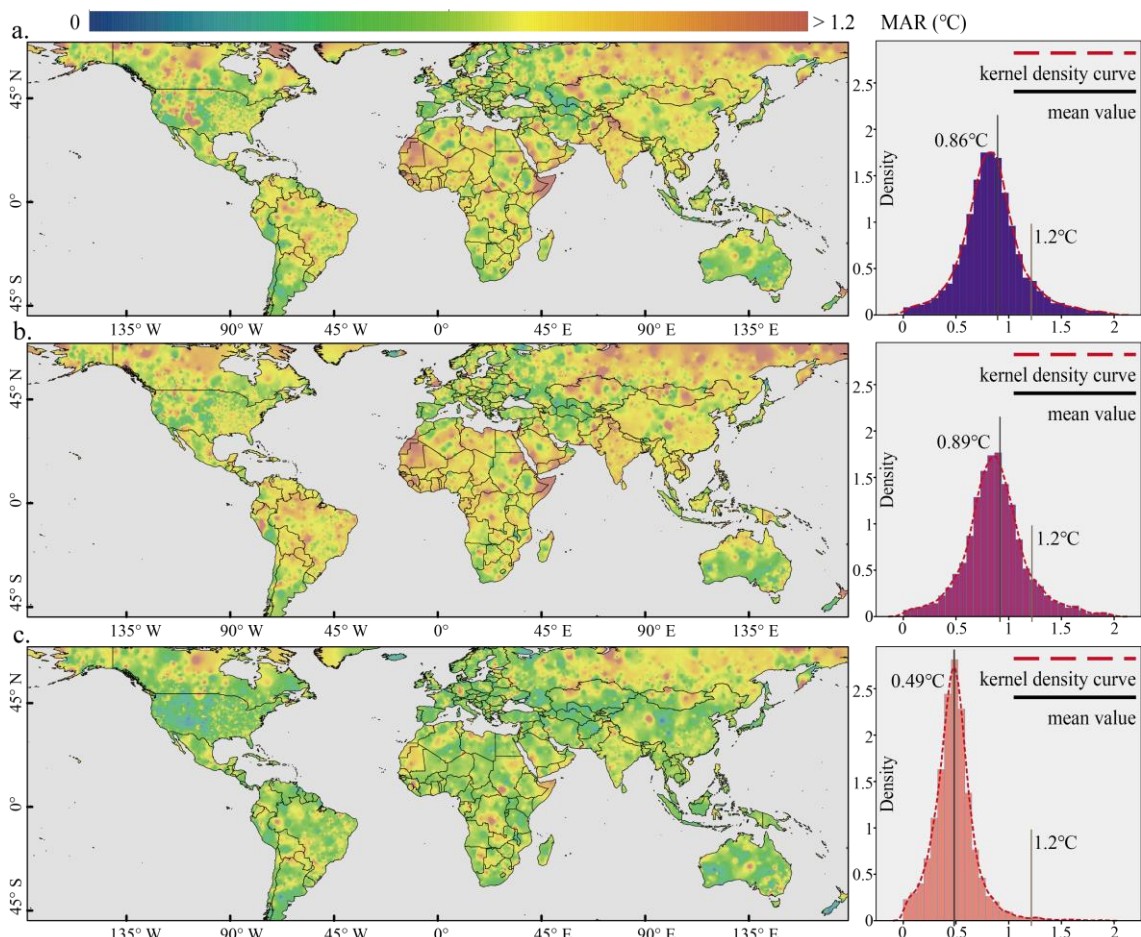

**Figure 4.** Spatially interpolated distribution and statistics of MAR between predicted UTCI and observed UTCU for global meteorological stations: (a) XGBoost; (b) LightGBM; (c) CatBoost.

**4.2 Spatial distribution of UTCI**

Based on the assessment of the three models, CatBoost emerged as the optimal choice for estimating the global monthly UTCI. Consequently, we employed CatBoost to produce the global monthly 1 km Universal Thermal Climate Index dataset spanning from 2000 to 2022, known as GloUTCI-M. Utilizing the monthly UTCI data for 2021, along with its annual


counterpart, we conducted an analysis to discern the spatial distribution pattern of the global UTCI (Fig. 5). Moreover, we extracted cold stress and thermal stress pixels for each winter month (refer to the nomenclature of the Northern Hemisphere, i.e., December, January, and February) as well as each summer month (refer to the nomenclature of the Northern Hemisphere, i.e., June, July, and August) in 2021, respectively. This enabled us to delineate the worldwide distribution of cold and thermal stress areas (Fig. 6).

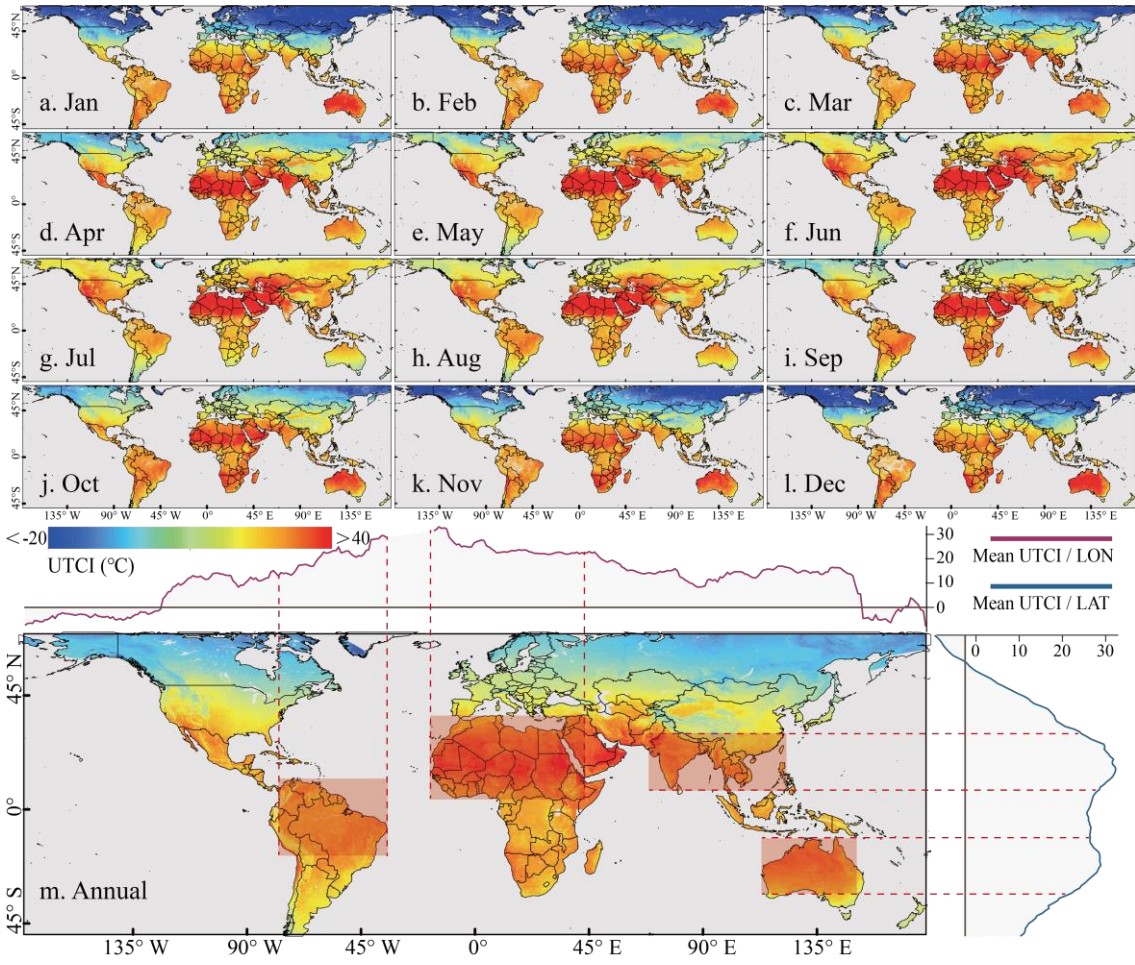

**Figure 5.** Spatial distribution of monthly and annual UTCI globally in 2021.

The distribution pattern of global monthly UTCI reveals notable differences between the Northern and Southern Hemispheres, as well as seasonal variations. In the Northern Hemisphere, there are prominent latitudinal variations, with UTCI generally decreasing from the equator towards higher latitudes across all months. The trend of UTCI in the Southern Hemisphere exhibits less significant changes with increasing latitude. Furthermore, the disparity in UTCI between the two hemispheres becomes more pronounced during the winter months, particularly in terms of latitude-related differences. Conversely, this difference diminishes during the summer months. The global distribution of monthly UTCI also exhibits

reasonable variations in accordance with the changing seasons. During the summer months (Figs. 5(f) - (g)), there is a
substantial concentration of high UTCI pixels (> 40°C), while low UTCI pixels (< -20°C) are less prevalent in both the
Northern and Southern Hemispheres. In contrast, the Northern Hemisphere experiences a notable abundance of low UTCI
pixels in the winter months (Figs. 5(a), (b), and (l)). Furthermore, we computed the mean UTCI at each longitude and
latitude based on annual UTCI data and represented them in a line plot (Fig. 5(m)). This plot highlights two regions with
relatively high UTCI in northern South America and northern Africa. In the latitudinal perspective, regions with relatively
high UTCI values are evident in southern Asia and Australia, respectively.

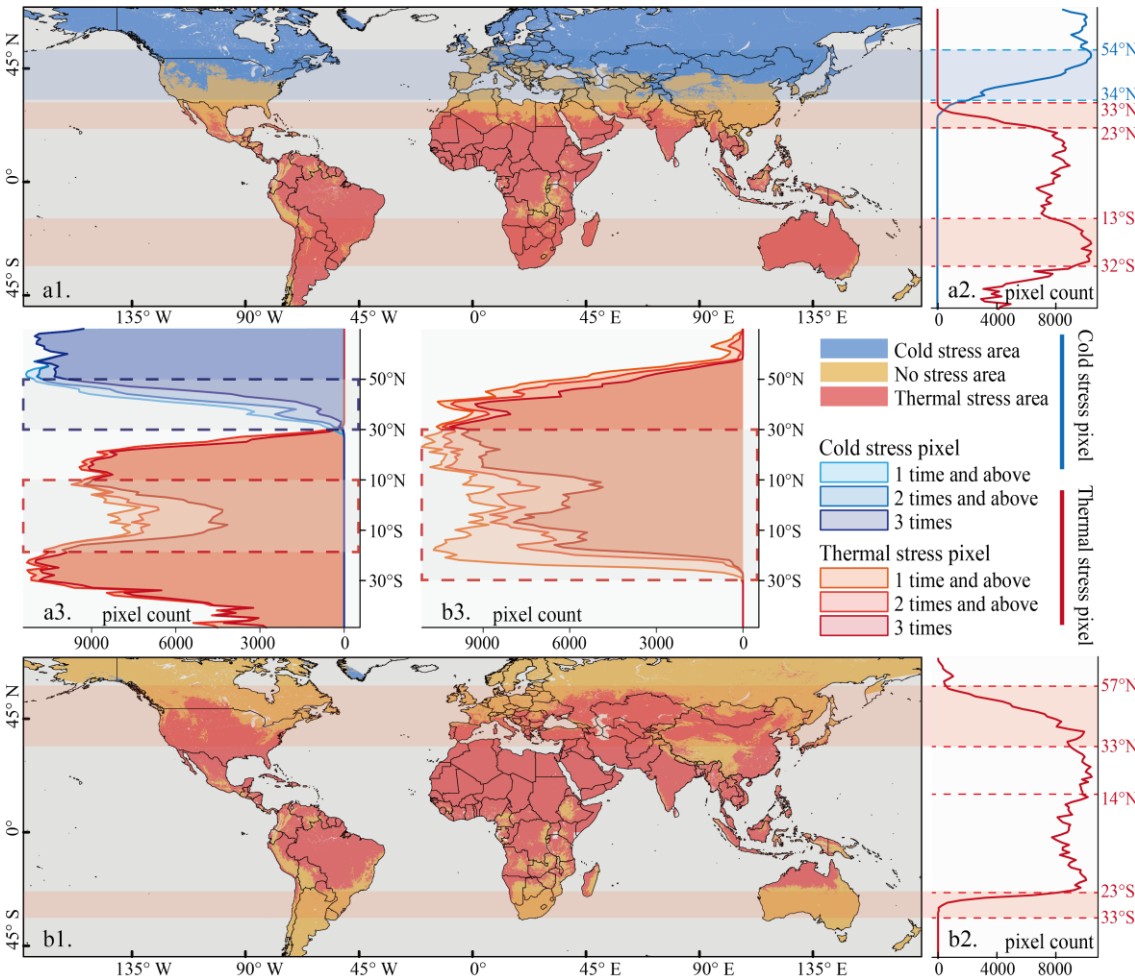

**Figure 6.** Distribution and statistics of cold and thermal stress areas globally in 2021: (a1) winter months; (b1) summer months; (a2)
latitudinal series of winter months; (b2) latitudinal series of summer months; (a3) type of pixel on the latitude series of winter months; (b3)
type of pixel on the latitude series of summer months.

The global distribution of cold stress areas (UTCI < 0°C) and thermal stress areas (UTCI > 26°C) during the winter
months, as determined by mean UTCI, exhibits significant latitudinal heterogeneity (Fig. 6(a1)). The concentration of cold
stress areas is particularly notable in the northern regions of North America and the Asian-European continent. Conversely,



no stress areas encompass regions such as the United States, the Mediterranean, and China. Thermal stress areas are widely dispersed, encompassing the majority of the southern hemisphere and the region between the equator and the Tropic of

Cancer. By employing BEAST analysis, we identified mutation points in the quantity of cold and thermal stress pixels along the latitudinal sequence (Fig. 6(a2)). Notably, the spatial range from 54°N to 34°N experiences a substantial reduction in cold stress pixels, while thermal stress pixels show a rapid increase between 33°N and 23°N, with another increase between 13°S and 32°S. Furthermore, the number of occurrences of cold and thermal stress pixels at each latitude during the three winter months was examined separately. It was observed that cold stress pixels in the region above 50°N remained relatively

stable throughout the three winter months, whereas there were significant monthly fluctuations in the region between 30°N and 50°N. Conversely, the thermal stress pixel displayed significant monthly fluctuations in the region of 15°S to 10°N (Fig. 6(a3)).

During the summer months, the global thermal stress area, characterized by a mean UTCI, exhibits widespread distribution, while the cold stress area is more sporadic (Fig. 6(b1)). The majority of the regions around the equator

(excluding the Tibetan Plateau) fall within the thermal stress area. Conversely, other regions, such as the northern part of North America, most of Europe, South America, Africa, and the southern part of Australia, are characterized by a high concentration of no stress areas. Given the limited presence of cold stress areas, we focused on plotting the number of thermal stress pixels occurrences along the latitudinal series and employed BEAST analysis to pinpoint mutation points (Fig. 6(b2)). The interval from 57°N to 33°N experiences a significant increase in the number of thermal stress pixel, and this

number remains consistently high within the 33°N-23°S interval (with fluctuations within the 14°N-23°S interval). However, a sharp decrease in the number of thermal stress pixel is observed in the 23°S-33°S interval. Furthermore, there are substantial monthly fluctuations in the number of thermal stress pixel occurrences across latitudes during the three summer months (Fig. 6(b3)). This phenomenon is particularly pronounced in the 30°N-30°S interval, where a large number of pixels experience thermal stress in only one or two of the summer months.

**4.3 Temporal trends of UTCI**

Utilizing GloUTCI-M, we examined the time-series evolution of global UTCI. Initially, we compiled monthly global UTCI from 2000 to 2022, along with mean annual UTCI from 2001 to 2021. Subsequently, we construct individual scatter plots to gain insight into the fluctuations in the global UTCI (Fig. 7). Furthermore, we extracted mean annual UTCI for global pixels and employed the Theil-Sen slope estimation method and the MK method to identify the trend of the global

UTCI (Fig. 8).

Between the years 2000 and 2022, there is a noteworthy increase in the mean global UTCI during the summer months (June - September) which is statistically significant ($p < 0.05$). In contrast, the trend in the mean global UTCI during winter is mostly non-significant ($p > 0.05$). Additionally, the trend in the mean annual global UTCI demonstrates a substantial elevation ($R^2 = 0.66,\ p < 0.05$) (Fig. 7 (m)). To gain a deeper understanding of how each month's UTCI

contributes to or suppresses the mean annual global UTCI from 2001 to 2021, we produced waterfall plots (Fig. 7 (n)). This



plot allows us to discern the positive or negative impact of each month's UTCI by comparing the difference between the annual UTCI and the UTCI for that specific month. The mean annual global UTCI for the period 2001-2021 is 17.24°C. Out of the twelve months, seven months (April - October) make a positive contribution towards achieving the mean annual global UTCI. Among these, July holds the most substantial contribution (+9.86°C), followed by August (+8.97°C).

Conversely, the remaining five months (January - March, November, and December) fall below the mean annual global UTCI. Among these, January exerts the most significant inhibitory effect (-11.96°C), while December and February also display notable inhibitory effects (-11.05°C and -9.33°C, respectively).

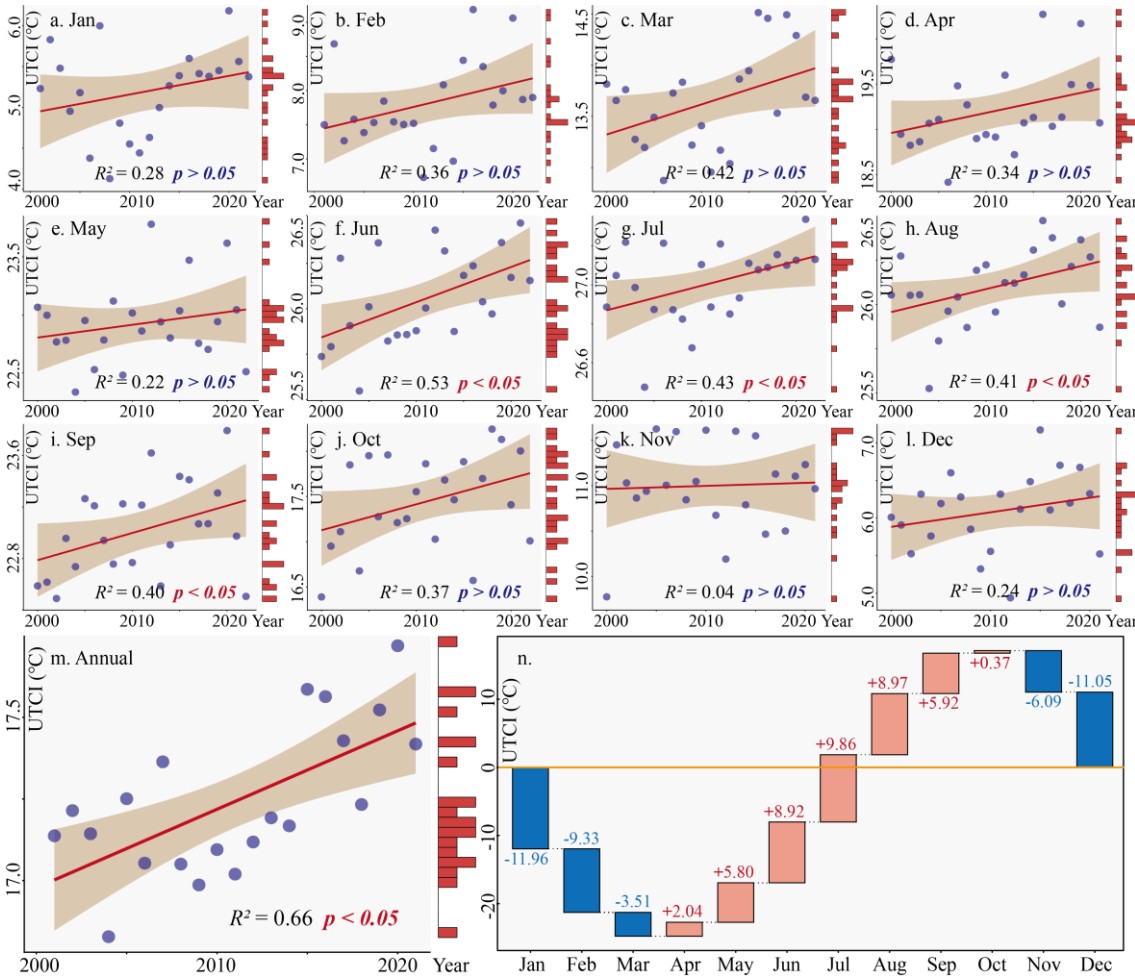

**Figure 7.** Global monthly and annual UTCI changes from 2000 to 2022: (a)-(l) monthly UTCI changes; (m) annual UTCI changes; (n)
difference between each monthly UTCI and the annual UTCI.

The annual UTCI trends for global pixels are predominantly characterized by increasing trends and stability, with fewer pixels exhibiting decreasing trends (Fig. 8 (a)). Regions displaying increasing UTCI trends include the western United States (Fig. 8 (b1)), the eastern part of Brazil (Fig. 8 (b2)), western Europe (Fig. 8 (b3)), Southeast Asia (Fig. 8 (b4)), and the eastern part of China (Fig. 8 (b5)). Regions with decreasing UTCI trends are primarily situated in southern Mexico (Fig. 8



(c1)), northern South America (Fig. 8 (c2)), central Africa (Fig. 8 (c3)), western India (Fig. 8 (c4)), and southwestern China
(Fig. 8 (c5)). Additionally, we conducted separate counts of the top 10 countries with the highest number of UTCI trends
showing increasing and decreasing patterns. Regions such as Russia in Asia and Europe, Brazil in South America, and Libya
in Africa have over 1,000,000 pixels exhibiting increasing UTCI trends, and these regions serve as hotspots driving the
increase in the mean annual global UTCI (Fig. 8 (d1)). Both China and India have more than 500,000 pixels displaying

decreasing UTCI trends, and these two countries play a significant role in mitigating the elevation of the mean annual global
UTCI (Fig. 8 (d2)).

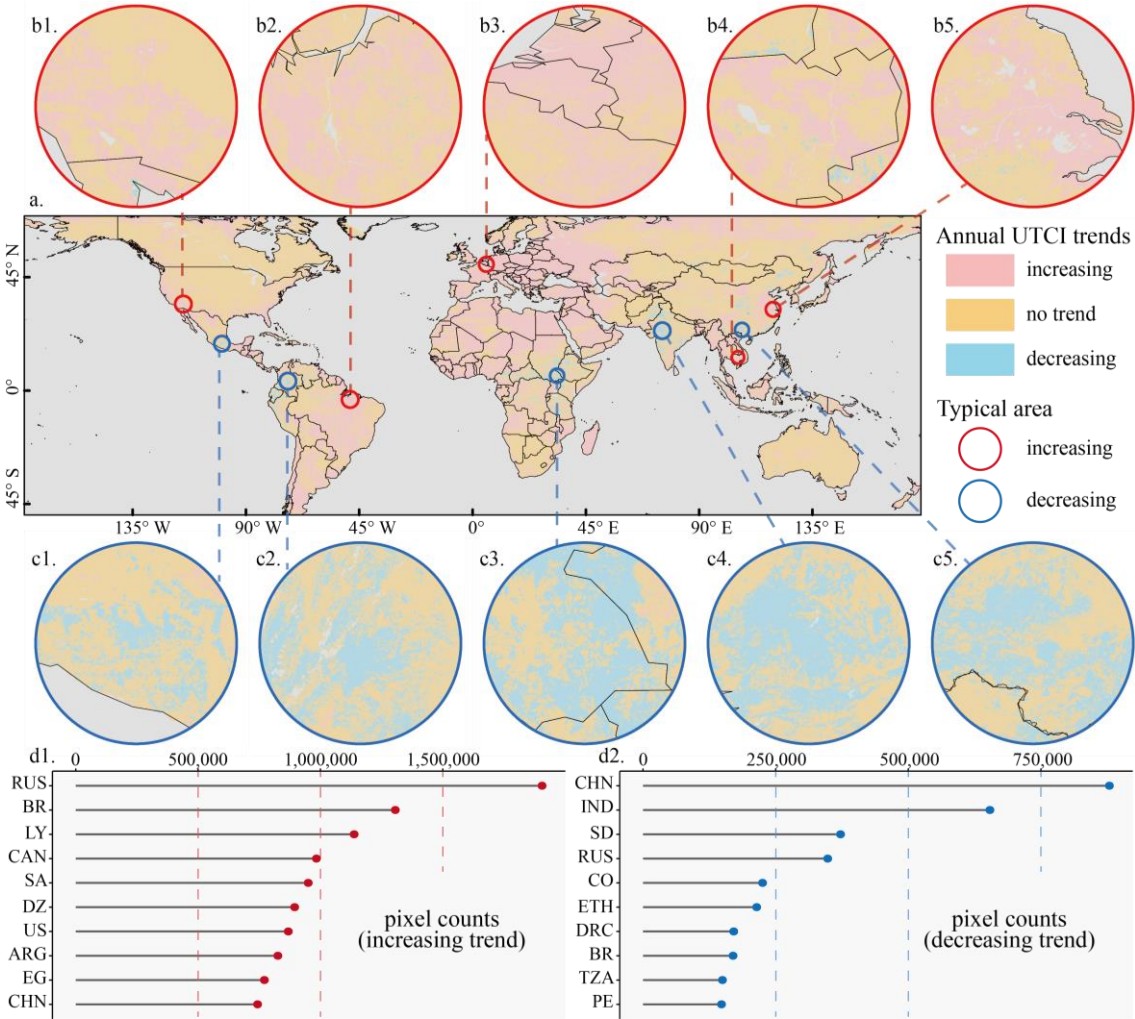

**Figure 8.** Trends in annual UTCI for global pixels: (a) spatial distribution; (b1) - (b5) typical areas with increasing trends in annual UTCI;
(c1) - (c5) typical areas with decreasing trends in annual UTCI; (d1) top 10 countries with the highest number of pixels showing an
increasing trend in the annual UTCI; (d2) top 10 countries with the highest number of pixels showing a decreasing trend in the annual
UTCI.



## 5 Discussion

### 5.1 Comparison with existing UTCI dataset

Given the availability of existing UTCI datasets, we undertake a comparative analysis of the spatial distributions of
ERA5-HEAT, HiTiSEA, and GloUTCI-M across various scales, including the intercontinental, city cluster, and city scales.
We extracted UTCI data from the ERA5-HEAT between 11:00 and 14:00 local time to calculate monthly UTCI for the year
2019. Simultaneously, we extracted the maximum daily UTCI from the HiTiSEA to derive monthly UTCI for the same year.
Moreover, we compared the data quality of the three types of datasets by plotting Taylor diagrams and using RMSE and $R^2$.
Specifically, using the year 2019 as a case study, we selected meteorological stations situated in East and South Asia as
representative samples. We then calculated the disparities between monthly UTCI obtained from the three datasets and those
derived from meteorological observation data for summer (June, July, and August), winter (December, January, and
February), as well as all twelve months throughout the year.

ERA5-HEAT, HiTiSEA, and GloUTCI-M manifest analogous UTCI spatial distribution patterns on the intercontinental
scale (Fig. 9). Nevertheless, at the city cluster and city scales, GloUTCI-M exhibits marked advantages. Specifically,
GloUTCI-M excels in delineating intricate variations in UTCI distribution within urban areas. Furthermore, we conducted a
comprehensive assessment by comparing the disparities between these three datasets and the observed UTCI collected from
meteorological stations. This evaluation, represented using Taylor diagrams, elucidates the disparities in accuracy among the
trio. When examining winter and year-round samples, all three datasets demonstrate a robust correlation with observed UTCI.
Notably, GloUTCI-M yields the smallest RMSE, with HiTiSEA following closely (Fig. 9 (d) and (f)). In the case of summer
month samples, the $R^2$ between GloUTCI-M and observed UTCI consistently exceed 0.95, whereas those between ERA5-
HEAT, HiTiSEA, and observed UTCI linger below 0.8. Furthermore, the RMSE between ERA5-HEAT and HiTiSEA in
comparison to observed UTCI significantly surpass those of GloUTCI-M ((Fig. 9 (e)). Hence, in contrast to ERA5-HEAT
and HiTiSEA, GloUTCI-M excels in portraying UTCI distribution at smaller spatial scales and attains superior data accuracy,
marked by stronger agreement with observed UTCI and diminished RMSE.



**Figure 9.** Comparison of spatial information and data accuracy among ERA5-HEAT, HiTiSEA and GloUTCI-M: (a) ERA5-HEAT at different spatial scales; (b) HiTiSEA at different spatial scales; (c) GloUTCI-M at different spatial scales; (d) comparison of three UTCI datasets with observed UTCI for winter months; (e) comparison of three UTCI datasets with observed UTCI for summer months; (f) comparison of three UTCI datasets with observed UTCI for all months.

## 5.2 Data availability at pixel scales

The existence of voids in the raster dataset of covariates, attributed to factors such as cloud cover and spatiotemporal discontinuities, results in the GloUTCI-M product lacking spatiotemporal seamlessness. To elucidate the global spatiotemporal availability of GloUTCI-M, we conducted a comprehensive assessment of pixel availability across 272 months within the GloUTCI-M. Spanning from 2000 to 2022, GloUTCI-M has a maximum missing pixel rate of 2.5%.



However, it's noteworthy that the majority of months within this time frame exhibit a missing pixel rate of less than 1%. Furthermore, when scrutinizing individual months, July and August consistently emerge with higher rates of missing pixel across all years than other months (Fig. 10 (a)).

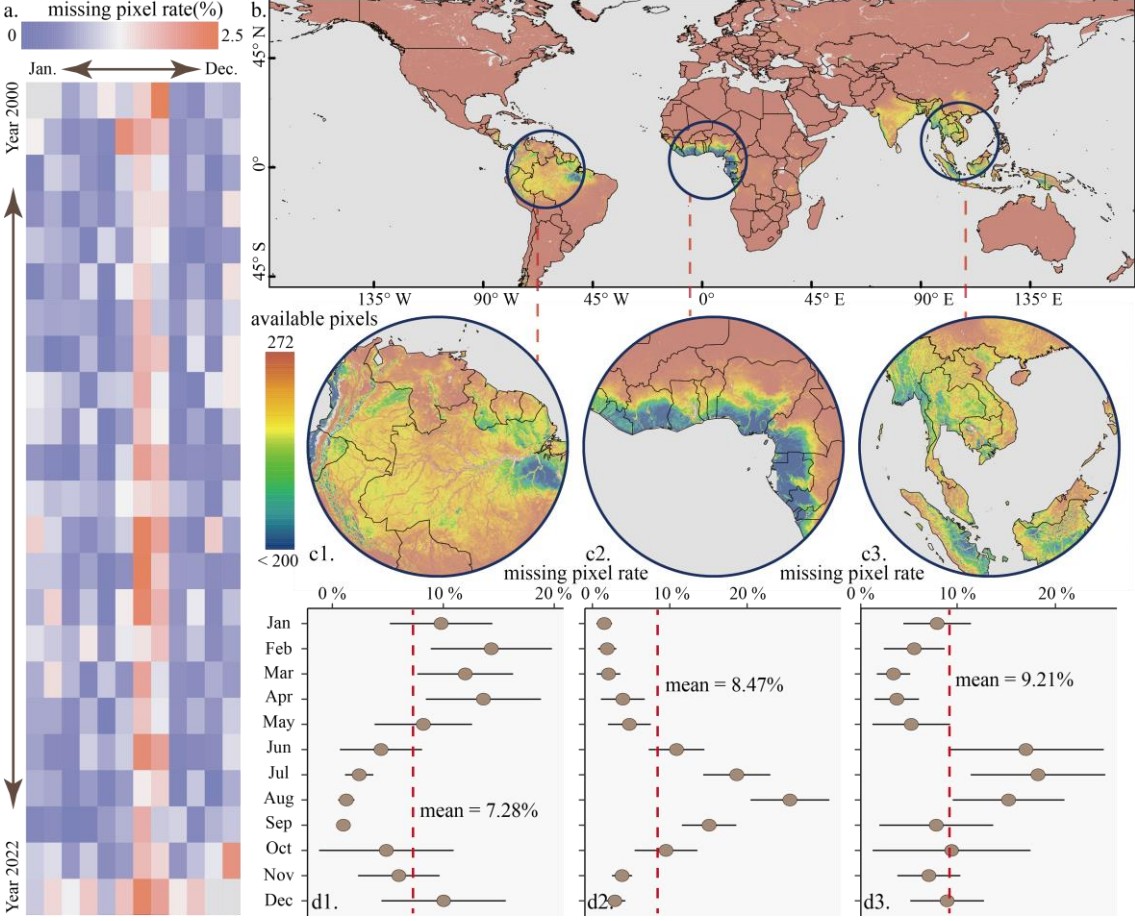

**Figure 10.** Available pixels for GloUTCI-M: (a) missing pixel for each month; (b) distribution of values of available pixels; (c1) - (c3)
areas of densely missing pixels; (d1) - (d3) missing pixel rate by month for areas of densely missing pixels.

GloUTCI-M boasts robust pixel availability across the majority of global regions (Fig. 10 (b)). Nevertheless, there are three regions characterized by densely missing pixels. These regions include northern South America (Fig. 10 (c1)), the western coastline of Africa (Fig. 10 (c2)), and the Southeast Asian territory (Fig. 10 (c3)). Examining the month-averaged missing pixel rates for these regions, we observe values below 10%. Southeast Asia reports the highest missing pixel rate at

9.21%, followed by the west coast of Africa at 8.47%, while northern South America records relatively lower rates at 7.28%. Notably, these regions exhibit varying trends in missing pixel rates by month. For northern South America, the period from January through May and December showcases above-average missing pixel rates, notably exceeding 10% from February through April (Fig. 10 (d1)). Conversely, the west coast of Africa experiences a concentration of missing pixels from June to



September, with August peaking at more than 20% (Fig. 10 (d2)). Southeast Asia exhibits heightened missing pixel rates
from June through August, all surpassing 15% (Fig. 10 (d3)).

### 5.3 Limitations and future works

The production of GloUTCI-M is intrinsically linked to the quality of the covariate data. Consequently, the existence of
missing pixels within these covariates results in varying degrees of gaps within the monthly UTCI raster of GloUTCI-M. To
enhance both the accuracy and data availability of the GloUTCI, a viable avenue is to identify and utilize spatiotemporally
seamless remote sensing data as novel covariates.

Several covariates, such as LST, LULC, and kNDVI, utilized in the GloUTCI-M production process are drawn from the
MODIS data. Consequently, GloUTCI-M exclusively comprises global monthly UTCI data spanning from March 2000 to
October 2022, as there is no access to MODIS data before 2000. A promising strategy for extending the temporal range of
the global monthly UTCI dataset involves leveraging available covariate data from different time periods to produce UTCI
datasets corresponding to those specific time periods. Subsequently, by fusing these datasets, the expansion of the temporal
coverage of the global monthly UTCI dataset.

### 6 Data availability

The GloUTCI-M comprises global monthly UTCI data at a spatial resolution of 1km, spanning from March 2000 to
October 2022. This dataset is openly accessible in GeoTIFF format via Zenodo: https://doi.org/10.5281/zenodo.8310513
(Yang et al., 2023). The dataset is expressed in degrees Celsius (°C) and is stored as an integer type (Int16). To utilize it
appropriately, one must divide the values by 100.

### 7 Conclusions

To address the existing gaps in UTCI data availability and enhance the applicability of UTCI in various domains, we
have produced a monthly UTCI dataset, GloUTCI-M, which boasts global coverage, a lengthy time series spanning from
March 2000 to October 2022, and a high spatial resolution of 1 km. GloUTCI-M is the result of amalgamating multiple data
sources (including LST, NTL, LULC, kNDVI, etc.) and employing an optimized machine learning model, CatBoost. Our
analysis of the spatial and temporal evolution of global UTCI, based on GloUTCI-M, reveals a rational distribution pattern
showcasing disparities between the northern and southern hemispheres, as well as seasonal fluctuations. Significant
latitudinal variations are apparent in the distribution of global cold and thermal stress areas. During the summer months
(June-September), the global mean UTCI experiences a notable increase, with an even more pronounced elevation observed
in the trend of the mean annual global UTCI. This trend, at the pixel level, is predominantly characterized by an increasing
trend, with fewer pixels displaying a decreasing trend. In the global UTCI trend, countries like Russia and Brazil emerge as
key contributors to the rising mean annual global UTCI, while countries such as China and India exert a greater influence in



mitigating this rise. In addition, when compared to existing UTCI datasets such as ERA5-HEAT and HiTiSEA, GloUTCI-M
excels in portraying UTCI distributions at fine spatial scales, offering superior data accuracy, stronger alignment with observed UTCI. We anticipate that the GloUTCI-M will serve as a valuable resource, providing comprehensive and accurate information to support research and policymaking across various domains, including meteorological science, health management, urban planning, and agriculture. Its utility extends to enhancing human comfort, reducing weather-related health risks, and facilitating better adaptation to the challenges posed by climate change.

**Author contributions.** ZY and JP designed the research and developed the methodology. ZY collected the data, conducted the analyses, and wrote the original manuscript. All authors reviewed and revised the manuscript.

**Competing interests.** The contact author has declared that none of the authors has any competing interests.

**Disclaimer.** Publisher's note: Copernicus Publications remains neutral with regard to jurisdictional claims in published maps and institutional affiliations.

**Financial support.** This research was financially supported by the National Natural Science Foundation of China (grant no. 42130505).

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
