# Peer review of "GloUTCI-M: A Global Monthly 1 km Universal Thermal Climate Index Dataset from 2000 to 2022"

_Earth System Science Data, 2023_

## Referee Comment (RC1)

This manuscript generated a global monthly 1 km universal thermal climate index dataset from 2000 to 2022. This work is meaningful and the result is basically satisfactory. However, some other problems in the manuscript are still concerned in the following:

1. The authors applied XGBoost series methods to generate the dataset. Why not use deep learning models?
2. More information on XGBoost, LightGBM and CatBoost should be exposed.
3. As stated in "Making the Earth clear at night: a high-resolution nighttime light image deblooming network", NTL data are subject to degraded issues. Did the authors preprocess NTL data?
4. More recent works are suggested to be included.

---

## Author Comment (AC1)

Dear Editor and Reviewers:

Thanks for your comments concerning our manuscript entitled "GloUTCI-M: A Global Monthly 1 km Universal Thermal Climate Index Dataset from 2000 to 2022". (No.: essd-2023-379). Those comments are all valuable and very helpful for revising and improving our paper, as well as the important guiding significance to our researches. We have studied comments carefully and have made correction which we hope meet with approval. The main corrections in the paper and the response to your comments are as follows:

**Reviewer #1:**

**Thanks for your comments on our paper. We have revised our paper according to your comments:**

This manuscript generated a global monthly 1 km universal thermal climate index dataset from 2000 to 2022. This work is meaningful and the result is basically satisfactory. However, some other problems in the manuscript are still concerned in the following:

**Response:**

Thank you for your nice comments on this research. We have revised the manuscript based on your comments and suggestions. These comments are valuable and helpful in revising and improving our paper, as well as providing important guidance for our research.

**01.** The authors applied XGBoost series methods to generate the dataset. Why not use deep learning models?

**Response:**

Thanks for your insightful question regarding our choice of using XGBoost series methods to generate the dataset instead of deep learning models. The decision to employ XGBoost series methods was based on several considerations specific to our research context.

Firstly, the nature of our dataset and the characteristics of the features make XGBoost series methods particularly well-suited. XGBoost series methods is known for its efficiency in handling tabular data, especially when dealing with a moderate-sized dataset with a relatively large number of features. It often outperforms deep

learning models in such scenarios. Furthermore, the training time and computational resources required for deep learning models can be substantial, especially considering the scale of our dataset. XGBoost series methods offer a good balance between model performance and computational efficiency, which is essential for our research objectives.

It's worth noting that we acknowledge the potential of deep learning models and their success in various domains. However, for the specific tasks addressed in our study, we found that XGBoost series methods align better with our objectives and constraints. Following your suggestion, we have elaborated in the manuscript the reasons for choosing to use machine learning models rather than deep learning models to generate the dataset.

**The following is the revised version:**

**(Lines 180-182)** We apply machine learning models to produce the UTCI dataset because they are more suitable for handling tabular data and provide a good balance between model performance and computational efficiency compared to methods such as deep learning models.

**02.** More information on XGBoost, LightGBM and CatBoost should be exposed.
**Response:**

Thanks for your request for more detailed information on XGBoost, LightGBM, and CatBoost. Below, I provide an expanded overview of each algorithm:

XGBoost: Known for its high performance and accuracy, XGBoost is particularly effective for tabular data. It excels in capturing complex relationships within the data and provides robust predictions. The regularization techniques in XGBoost help prevent overfitting, making it suitable for our dataset.

LightGBM: LightGBM is designed for efficiency and scalability. It uses a histogram-based approach for tree construction, which accelerates the training process and makes it well-suited for large datasets. Its ability to handle categorical features efficiently is advantageous for our diverse feature set.

CatBoost: CatBoost is specifically designed to handle categorical features without the need for extensive preprocessing. This makes it convenient for our dataset, which contain a mix of categorical and numerical features. CatBoost's categorical boosting approach contributes to its robust performance.

All three algorithms (XGBoost, LightGBM, and CatBoost) are well-documented, widely used, and supported by a large community. This ensures ease of implementation and troubleshooting, facilitating a smoother integration into our research.

**The following is the revised version:**

**(Lines 188-191)** XGBoost is particularly effective for tabular data. It excels in capturing complex relationships within the data and provides robust predictions. It excels as a tool for massively parallel boosting trees and is characterized by its efficiency, flexibility, and portability.

**(Lines 203-206)** It uses a histogram-based approach for tree construction, which accelerates the training process and makes it well-suited for large datasets. Its ability to handle categorical features efficiently is advantageous for diverse feature set. LightGBM utilizes a leaf-wise tree growth strategy to select the leaf node with the highest gain at each split, enabling faster, deeper tree growth and improving model accuracy.

**(Lines 213-216)** CatBoost is specifically designed to handle categorical features without the need for extensive pre-processing. It computes statistics on categorical features, such as category frequency, and uses hyperparameters to generate new numerical features. CatBoost's categorical boosting approach contributes to its robust performance.

**03.** As stated in "Making the Earth clear at night: a high-resolution nighttime light image deblooming network", NTL data are subject to degraded issues. Did the authors preprocess NTL data?

**Response:**

Thank you for bringing attention to the potential degraded issues in the NTL data. We selected the NPP-VIIRS-like NTL data as the NTL data used. NPP-VIIRS-like NTL data have an excellent spatial pattern and temporal consistency which are similar to the composited NPP-VIIRS NTL data.

The proposed cross-sensor calibration is unique due to the image enhancement by using a vegetation index and an auto-encoder model. The production process of the NPP-VIIRS-like NTL data has implemented a series of preprocessing steps (Noise Reduction, Radiometric Calibration, Spatial Resolution Enhancement, Temporal Filtering), which can improve the resolution of potential degradation issues in NTL data. We have supplemented the revised manuscript with a more detailed description of the NPP-VIIRS-like NTL data.

**The following is the revised version:**

**(Lines 114-118)** NTL is indicative of human activities and urbanization. We utilized NPP-VIIRS-like NTL data, available at a spatial resolution of 500 m. This dataset effectively combines data from two NTL sources (DMSP-OLS and NPP-VIIRS), extending the temporal range of NTL observations (Chen et al., 2021). In response to the potential degradation of NTL data (Bai et al., 2023), a series of pre-processing steps

in the production of NPP-VIIRS-like NTL data and the proposed cross-sensor calibration can effectively improve the problem.

**04.** More recent works are suggested to be included

**Response:**

Thanks for your valuable suggestion to include more recent works in our manuscript. In response to your suggestion, we have conducted a literature review to identify and incorporate relevant and recent works that contribute to the advancements in our research topic.

---

## Author Comment (AC2)

Dear Editor and Reviewers:

Thanks for your comments concerning our manuscript entitled "GloUTCI-M: A Global Monthly 1 km Universal Thermal Climate Index Dataset from 2000 to 2022". (No.: essd-2023-379). Those comments are all valuable and very helpful for revising and improving our paper, as well as the important guiding significance to our researches. We have studied comments carefully and have made correction which we hope meet with approval. The main corrections in the paper and the response to your comments are as follows:

**Reviewer #2:**

**Thanks for your comments on our paper. We have revised our paper according to your comments:**

The Universal Thermal Climate Index (UTCI), an important approach to human comfort assessment, plays a pivotal role in gauging how the human adapts to meteorological conditions and copes with thermal and cold stress. The study developed an interesting Global Monthly 1 km Universal Thermal Climate Index Dataset from 2000 to 2022. The study's structure and analysis are good, and the limitations and uncertainty are discussed. I like the study and there are no more comments.

**Response:**

Thanks for your nice comments on this research. According to your suggestions, we have improved the introduction and methodology section in the revised manuscript and corrected several errors in the previous draft.

**01.** I only have a suggestion in the introduction part. you can describe more details to state your argument to develop this dataset.

**Response:**

Thanks for your suggestion regarding the introduction section of our manuscript. In response to your suggestion, we enhanced the introduction section by including more specific details to articulate the significance of developing this dataset. We expanded on the following aspects:

Motivation: UTCI can better characterize the thermal and cold stresses experienced by humans. However, the quantity and quality of UTCI datasets are insufficient, which hinders in-depth research and the application of UTCI. To facilitate

the widespread future applications of UTCI data, we have produced GloUTCI-M, a monthly UTCI dataset characterized by global coverage, a long-time series, and high spatial resolution.

Significance: A suite of globally accessible, long-time series and high spatial resolution UTCI datasets can enhance the precision and practicability of UTCI for urban and landscape scale investigations.

**The following is the revised version:**

**(Lines 55-57)** The UTCI, with its incorporation of multiple meteorological variables and hallmark objectivity and comprehensiveness, can better characterize the thermal and cold stresses experienced by humans.

**(Lines 59-67)** However, the quantity and quality of UTCI datasets are insufficient, which hinders in-depth research and the application of UTCI. The existing UTCI datasets predominantly exhibit low spatial resolutions, such as the ERA5-HEAT with a spatial granularity of 0.25° (encompassing the globe) and the HiTiSAE with a spatial granularity of 0.1° (encompassing East and South Asia) (Di Napoli et al., 2021; Yan et al., 2021). These prevailing UTCI datasets often inadequate for urban and landscape scale investigations, given that these studies necessitate data of higher spatial resolution to accurately capture intra-urban meteorological variations and human perceptions (Peng et al., 2021; Yang et al., 2021; Cao et al., 2022). Therefore, the development of a suite of globally accessible, long-time series and high spatial resolution UTCI datasets is imperative. This initiative will address the existing void in UTCI data availability and enhance the precision and practicability of UTCI for urban and landscape scale investigations.

**02.** Explain more about why you use XGBoost, LightGBM, and CatBoost.
**Response:**

Thanks for your question regarding the choice of XGBoost, LightGBM, and CatBoost in our study. The selection of these machine learning models was made based on careful consideration of several factors that align with our research objectives and dataset characteristics.

Firstly, the nature of our dataset and the characteristics of the features make machine learning models particularly well-suited. machine learning model is known for its efficiency in handling tabular data, especially when dealing with a moderate-sized dataset with a relatively large number of features. Furthermore, the training time and computational resources required for deep learning models can be substantial, especially considering the scale of our dataset. machine learning models offer a good balance between model performance and computational efficiency, which is essential for our research objectives.

XGBoost: Known for its high performance and accuracy, XGBoost is particularly effective for tabular data. It excels in capturing complex relationships within the data and provides robust predictions. The regularization techniques in XGBoost help prevent overfitting, making it suitable for our dataset.

LightGBM: LightGBM is designed for efficiency and scalability. It uses a histogram-based approach for tree construction, which accelerates the training process and makes it well-suited for large datasets. Its ability to handle categorical features efficiently is advantageous for our diverse feature set.

CatBoost: CatBoost is specifically designed to handle categorical features without the need for extensive preprocessing. This makes it convenient for our dataset, which contain a mix of categorical and numerical features. CatBoost's categorical boosting approach contributes to its robust performance.

All three algorithms (XGBoost, LightGBM, and CatBoost) are well-documented, widely used, and supported by a large community. This ensures ease of implementation and troubleshooting, facilitating a smoother integration into our research.

**The following is the revised version:**

(**Lines 188-191**) XGBoost is particularly effective for tabular data. It excels in capturing complex relationships within the data and provides robust predictions. It excels as a tool for massively parallel boosting trees and is characterized by its efficiency, flexibility, and portability.

(**Lines 203-206**) It uses a histogram-based approach for tree construction, which accelerates the training process and makes it well-suited for large datasets. Its ability to handle categorical features efficiently is advantageous for diverse feature set. LightGBM utilizes a leaf-wise tree growth strategy to select the leaf node with the highest gain at each split, enabling faster, deeper tree growth and improving model accuracy.

(**Lines 213-216**) CatBoost is specifically designed to handle categorical features without the need for extensive pre-processing. It computes statistics on categorical features, such as category frequency, and uses hyperparameters to generate new numerical features. CatBoost's categorical boosting approach contributes to its robust performance.

**03.** Why China and India exert a more inhibitory influence on this trend?
**Response:**

Thanks for your insightful question. In our study, we found that more than 500,000 pixels in both China and India show a decreasing trend in UTCI. This number

is much larger than other countries. Therefore, these two countries play an important role in mitigating the elevation of the mean annual global UTCI.

An article showing that China and India dominate the global greening of vegetation may explain this (Chen, C., Park, T., Wang, X. et al. China and India lead in greening of the world through land-use management. *Nat Sustain*, 122–129 (2019). https://doi.org/10.1038/s41893-019-0220-7). However, a more accurate and valuable analysis of the cause needs to be followed up with more in-depth research.

**The following is the revised version:**

 **(Lines 395-397)** Both China and India have more than 500,000 pixels displaying decreasing UTCI trends, and these two countries play a significant role in mitigating the elevation of the mean annual global UTCI.